# Seismic Tests of Full Scale Reinforced Concrete T Joints with Light External Continuous Composite Rope Strengthening—Joint Deterioration and Failure Assessment

**DOI:** 10.3390/ma16072718

**Published:** 2023-03-29

**Authors:** Martha Karabini, Theodoros Rousakis, Emmanouil Golias, Chris Karayannis

**Affiliations:** 1Laboratory of Reinforced Concrete and Seismic Design of Structures, Democritus University of Thrace, 67100 Xanthi, Greece; mkarampi@civil.duth.gr (M.K.); egkolias@civil.duth.gr (E.G.); 2Laboratory of Reinforced Concrete and Masonry Structures, Aristotle University of Thessaloniki, 54636 Thessaloniki, Greece; karayannis@civil.auth.gr

**Keywords:** NSM strengthening, bonded carbon FRP ropes, cyclic loading, joints, reinforced concrete structures

## Abstract

Beam–column connections (joints) are one of the most critical elements which govern the overall seismic behavior of reinforced concrete (RC) structures. Especially in buildings designed according to previous generation codes, joints are often encountered with insufficient transverse reinforcement detailing, or even with no stirrups, leading to brittle failure. Therefore, externally bonded composite materials may be applied, due to the ease of application, low specific weight and corrosion-free properties. The present work assesses the seismic performance of insufficiently reinforced large-scale T beam–column connections with large and heavily reinforced beams. The joints receive externally bonded NSM X-shaped composite ropes with improved versatile continuous detailing. The columns are subjected to low normalized axial load, while the free end of the beam is subjected to transverse displacement reversals. Different failure criteria are investigated, based on the beam free-end transverse load, as well as on the joint region shear deformations, to critically assess the structural performance of the subsystem. The experimental investigation concludes that cyclic loading has a detrimental effect on the performance of the joint. Absence of an internal steel stirrup leads to earlier deterioration of the joint. The unstrengthened specimens disintegrate at 2% drift, which corresponds to 34 mm beam-end displacement, and shear deformation of the joint equal to 30 × 10^−4^ rad. The composite strengthening, increases the structural performance of the joint up to 4% drift which corresponds to 68 mm of beam-end displacement and shear deformation of the joint equal to 10 × 10^−4^ rad. The investigated cases of inadequate existing transverse reinforcement in the joint and light external FRP strengthening provide a unique insight into the required retrofits to achieve different levels of post-yielding displacement ductility under seismic loading at 2%, 3% and 4% drift. It allows for future analytical refinements toward reliable redesign analytical models.

## 1. Introduction

The mechanical performance of reinforced concrete (RC) beam–column connection subsystems is very critical for the overall seismic-resistant performance of the structure. the ideal desired response is for the RC joint to remain in the elastic region and plastic hinges to form at both ends of the beams. Therefore, protection of the RC joints against detrimental damage accumulation is most crucial, together with the protection of the columns. Local retrofit of beam–column connections can be employed in many cases to avoid costly and time-consuming global interventions [1,2,3,4,5]. Some of the well-known retrofitting techniques of RC joints include RC jacketing, steel plate or steel rod jacketing, and advanced shear strengthening with fiber-reinforced polymer (FRP) sheets [6,7,8].

Externally-bonded composite materials have been extensively used in different configurations in strengthening of RC columns and beams as well as in joints [9,10,11,12,13,14,15,16]. FRPs, or even SRPs [17], have been used as a lightweight, flexible and corrosion-resistant local retrofitting alternative to traditional RC or steel jacketing, especially in T-shaped RC joints [18,19,20,21,22,23,24], as well as in beams and columns [25,26,27,28,29,30].

In related past work, FRP sheets in different configurations have been externally bonded onto full scale RC joints to provide reinforcing fibers in critical directions and ensure increased shear strength and efficient FRP sheet anchorage, even with the use of additional steel plates in some cases [11,18,31,32]. In general, based on the obtained results, FRP strengthening succeeded in improving the structural behavior of the joints. However, the issue of the presence of a transverse beam (or beams) and slab jeopardizes the efficiency of most of the proposed FRP sheet detailing. Similarly, scaled specimens demonstrated the efficiency of seismic strengthening of joints or explored the shear capacity of the joint region when receiving shear FRP strengthening [7,10,19]. 

Recently, flexible composite rope strengthening has offered some unique application advantages as it allows for external wrapping of columns even without the use of impregnation or bonding agents. It is self- anchored, may introduce fire-resistant basalt fibers and presents non-fracture and strain redistribution characteristics prolonging the axial strain ductility of the RC column and protecting it against collapse [25,33,34]. Carbon FRP ropes with resin impregnation were successfully applied as near-surface mounted (NSM) or as embedded through section (ETS) in joints [35]. Resin-impregnated FRP ropes have been used as flexural or shear strengthening reinforcement in beams [36]. 

Full scale RC beam column joints subjected to cyclic loading and comparing different FRP rehabilitation methods, such as CFRP sheets and CFRP ropes placed in an X shape in the joint region alone, or in combination with beam or/and column retrofit, have been investigated [37,38]. The external CFRP rope technique has revealed in all cases the versatility and suitability for demanding retrofits of RC joints, being embedded inside grooves and being able to go through transverse RC members if needed.

This study reports the test results from critical cases of T-shaped large-scale deficient RC beam column connection subsystems with large and heavily reinforced beams. The columns receive low normalized axial load, while the free end of the beam is subjected to transverse displacement reversals. The cyclic displacement levels are gradually increased in each loading cycle, following characteristic values of beam drifts before and after the steel yielding of the beam reinforcement. Each displacement level includes three steps of identical drift. The three repetitions of cyclic displacements aim to explore the effects of deterioration of shear capacity of joint during seismic excitation. Different failure criteria are investigated based on the beam free-end transverse load, as well as on the joint region shear deformations, to critically assess the structural performance of the subsystem. The failure is based (a) on significant disintegration of the joint region deviating from the elastic response, and (b) on non-recoverable damage of the beam–column subsystem considered below the 0.8 P_max_. The experimental investigation concludes that cyclic loading has a detrimental effect on the performance of the joint. Absence of an internal steel stirrup leads to earlier deterioration of the joint. Suitably designed light strengthening with NSM X-shaped composite ropes may upgrade remarkably the ductility of the beam–column connection, delaying the disintegration of the joint and achieving higher drift levels at failure. The carbon FRP rope follows an improved continuous versatile detailing that may allow for more layers of rope inside the grooves. The experimental results may serve for future development of redesign tools. 

## 2. Specimens’ Dimensions, Reinforcement Detailing and Material Properties

### 2.1. Main Characteristics of As-Built Specimens

The tests investigate the performance of two-dimensional T beam–column connection subsystems consisting of half-length upper and bottom columns connected with half-length beams and their shared joint region. The total length of the two half-columns is 2.95 m (including 0.5 m of vertical joint size) and represents typical floor height. The clear half span of the beam is 1.875 m, while of the column is 1.225 m. The cross-section of the column has dimensions 300 mm × 400 mm and the beam 250 mm × 500 mm, respectively. These cross-section dimensions form a large joint volume element at the connection of the three ‘beam’ elements with dimensions 400 mm × 500 mm × 250 mm. Further, the columns have eight bars of 14 mm diameter (reinforcement with number (4) in Figure 1) and an 8 mm diameter closed stirrup (reinforcement with number (3) in Figure 1). The beam has eight longitudinal bars of 16 mm diameter (reinforcement with number (2) in Figure 1), symmetrically placed four at the bottom and four at the top of the section, and Φ8 closed stirrup (reinforcement with number (1) in Figure 1). T beam–column CON0 has no stirrup in the joint region, while CON1 has one stirrup. The exact detailing of the reinforcements, the anchorage detailing of the Φ16 bars in the joint region, as well as the position of the pinned supports for the columns and of the pinned actuator at the free end of the beam, are presented in Figure 1.

The maximum horizontal shear that can be induced in the joint by the beam’s tensile reinforcement (4 Φ 16 mm) is as high as V_jhd_ = 377 kN and the corresponding shear stress is τ = 3.3 MPa, as considered in Eurocode 8 part 1 or Greek Retrofit Code [39]. According to ACI 318 [40], external joints have to satisfy the relationships ΣΜR_c_/MR_b_ > 1.40 and φV_n_ > V_u_, where φ is the strength reduction factor, V_n_ the nominal shear strength and V_u_ the maximum required value of shear. Since ΣMR_c_/MR_b_ = 1.4, the development of a plastic hinge is expected within the critical region of the beam.

The above-mentioned detailing was based on the models by Greek Retrofit Code (KANEPE) [41] as well as on the well-established model introduced by Tsonos [6,42]. The Tsonos model determines the ultimate shear τ_ult_ (and γ_ult_ = τ_ult_/ f_c_^0.5^). Then, the factor γ_ult_ is compared to the developed shear τ_cal_ (and γ_cal_ = τ_cal_/f_c_^0.5^). The values of γ_cal_ are little less than the corresponding values of the ultimate shear cracking γ_ult_ for joints with and without a stirrup inside the joint region. Therefore, it is deduced that the cracking system is expected to be developed both in the beam and the joint body after the yielding of beam tensile reinforcement. These predictions are experimentally verified as it is presented in the subsequent sections. Similarly, the KANEPE model suggests there will be diagonal tensile cracking inside the joint region in both cases, i.e., with or without a stirrup in the joint region.

As already mentioned, the focus of this paper is on the effects of low internal steel stirrup quantity or of light external CFRP rope strengthening in order to estimate low bound adequate joint strengthening. Therefore, except for the CON0 and CON1 specimens, another two counterparts, namely CON0F2X and CON1F2X received innovative and versatile X-shaped external near-surface mounted (NSM) strengthening with flexible carbon FRP ropes with improved detailing.

### 2.2. Concrete and Steel

Ready mixed concrete was used to cast all four reinforced concrete specimens with the same batch without any construction joints. The concrete compressive strength was f_c_′ = 22.4 MPa, based on the mean value of three standard cylinders with diameter of 150 mm and height 300 mm, tested at 28 days. The compression machine and gathered results are presented in Figure 2. The steel used for the construction of the cages of the internal longitudinal bars and stirrups was of quality B500C, suitable for seismic-resistant structures. The tensile stress at yielding of the steel was equal to 550 MPa and the tensile strength (at ultimate) was 650 MPa.

### 2.3. Carbon FRP Flexible Ropes and Detailing of NSM Strengthening

The characteristics of the carbon FRP ropes used for the strengthening of the CON0F2X and CON1F2X joints are given by the manufacturer based on the net fiber content [43] (SikaWrap FX-50C, 2017, Sika Hellas SA, Athens, Greece): the tensile strength equals 4000 MPa, the tensile modulus of elasticity equals 240 GPa and the cross-section A_f_ of the carbon fibers of the used ropes A_f_ > 28 mm^2^. It corresponds to an ultimate force of 50 kN/rope cross-section. The carbon FRP rope is delivered already resin-impregnated per carbon filaments by the manufacturer and, therefore, favors easy handling. That is, the composite rope is flexible enough to follow the geometry of the concrete surface without damaging the carbon fibers during the process of the in-situ impregnation of the rope with the adhesive or during stretching of the rope against the concrete surface or against multiple rope rounds. The compressive strength of the adhesive resin for the in-situ application is 34 MPa, its tensile strength in flexure is 41 MPa, the tensile strength is 24 MPa whereas the tensile adhesion strength is given by the manufacturer as >4 MPa. Finally, the compressive strength of the resin paste after seven days is almost 114 MPa [43,44].

The flexible composite rope is applied inside grooves with a depth of 25 mm and a width of 50 mm. While the grooves decrease the shear strength of the joint, their application is necessary, as the X shape of the rope tends to move when it comes to tension. The notches that are on the width of the beams, at least, are necessary, so that the rope does not leave its initial position. Therefore, it was decided to provide adequate grooving all around the perimeter of the region under retrofit. An adequate concrete cover has to be provided, as the internal steel reinforcement has to be protected against direct damage or galvanic corrosion from contact with carbon fibers. The grooves follow the diagonals of the joint at both faces (see also [45]). The grooves terminate in horizontal segments outside the height of the joint, inside the reinforced concrete columns sections, as depicted in Figure 3. The grooves are of great importance as they prevent the rope from slipping over the back surface of the column. They are well rounded at the eight corners of the grooves in the columns to maximize the force exerted by the ropes against the two diagonal concrete sections and to avoid excessive stress concentration that may cause damage. The dust is then cleaned with compressed air. The second step is the application of a two-component epoxy resin in the notches as an adhesive substance between the ropes and the concrete. Instead of cutting the rope in two different layers (see [35,37,38,46,47]), here the continuous flexible rope is cut for the total length of the two layers and for the self-anchorage length and is resin-impregnated. The anchorage of the rope started at the point where the two diagonals intersect and was anchored after two laps and at the corresponding intersection point of the opposite face of the joint. The improved detailing ensures higher versatility in the application of the strengthening and the need for lower dimensions of the grooves in case of higher number of layers. The composite rope is placed inside the diagonal grooves by hand while stretching it against the concrete member. Finally, the grooves are fully filled with the epoxy resin paste (Figure 3). Thorough filling of the notches with high strength epoxy paste ensures the rehabilitation of the extracted concrete cover of the notches.

## 3. Test Setup and Loading Protocol

The test setup is presented in Figure 4. The T specimen is rotated by 90° so that the columns are in the horizontal direction whereas the beam is in the vertical direction. The end supports of the columns are hinged to allow rotation and simulate the inflection points of the columns in the middle of the floors in a real seismic-resistant framed structure. Both columns are subjected to constant low-magnitude concentric axial compressive load equal to N_c_ = 0.05 A_c_f_c_ =134 kN throughout the testing procedure.

All beams are subjected to fully symmetric transverse cyclic deformation near their free end, at a length of 1.475 m, by a swivel connector with the actuator (depicted in the horizontal direction in Figure 4). The T specimens were subjected to eight successive steps of increasing displacement with corresponding drift ratios equal to 0.50% (8.5 mm), 0.75% (12.75 mm), 1.0% (17 mm), 1.5% (25.5 mm), 2% (34.0 mm), 3% (51 mm), 4% (68 mm) and 5% (85 mm), respectively. Each loading step included three full displacement cycles, as shown in Figure 5, to assess the deterioration of the shear capacity of the joint and the low-cycle fatigue of steel. The imposed load was measured by a load cell, while the displacements of the column, the beam, and the joint area were measured by a linear variable differential transducer. The recorded results involve transverse beam load (P) and beam end displacement (δ) as well as diagonal deformations (γ) of the joint throughout testing for constant column axial load (N).

## 4. Experimental Test Results

The P-δ hysteretic curves for the four specimens are presented in Figure 6. It can be concluded that in terms of load carrying capacity, all joints with or without a steel stirrup in the joint region and with or without CFRP rope strengthening, develop the full load that corresponds to the yielding of the tensile steel reinforcement of the beam for both push and pull directions in a symmetric way. This is observed at 1% drift for the unstrengthened specimens and at 1.5% for the strengthened ones. As per the loading cycle, joints CON1F2X or CON0F2X externally strengthened with CFRP ropes exhibited an improved hysteretic response in comparison to the unstrengthened counterpart specimens CON1 and CON0. This is particularly pronounced in the high loading drifts of 3% and 4% or the loading cycles of the 6th and 7th loading steps (Figure 6). Similarly, joints CON1 or CON1F2X with the addition of a steel stirrup in the joint region revealed a better hysteretic response than the corresponding joints CON0 and CON0F2X without a joint steel stirrup. What is more interesting is that the comparison of P-δ behavior between CON1 and CON0F2X suggest that the two specimens present a rather equivalent mechanical response. Structural failure of the member may be defined at the drift level in which the post-peak transverse load of the beam reduces to 80% of the peak one, i.e., 0.8 P_max_ (see [39,41,48]). Therefore, Figure 6 suggests that specimen CON0 fails at a bearing load of 104.5 kN during the 2nd reversal at step 5 which corresponds to 2% drift. Specimen CON1 fails at 106.4 kN load during the 2nd reversal of the 6th step corresponding to 3% drift. Similarly, specimen CON0F2X fails at 105 kN during the 2nd cycle at 3% drift. Finally, CON1F2X fails at 107.4 kN during the 3rd cycle of step 6 at 3% drift. However, the bearing load of the 1st cycle at 4% drift is at 0.76 P_max_ which is reasonably close to 0.8 P_max_ limit for CON1F2X. The above results, for all the specimens, are similar and symmetrical for both push and pull directions.

As a result of the design target, the severe damage in specimen CON0 is expected to be located mainly at the joint body after yielding of the tensile steel of the bars of the beam. Actually, from the first cycles of loading, the cracks were concentrated at the region of the joint, and the subsequent loading cycles resulted in a gradual increase in the width of the cracks across the joint region. Finally, the concrete of this region was partially fragmented. The damage state of the specimen CON0 at the 5th step of the loading (drift 2%) is presented in Figure 7a. It is observed that X-shaped cracks have been formed in the joint body and have increased as the steps progressed. The failure of the joint is presented in Figure 7b, where the damage state of the specimen at the end of the loading procedure (at the 7th step—drift 4%) is shown. The damage state of the specimen CON1 at the 5th step of the loading (drift 2%) is presented in Figure 7c. Similarly, CON1 reveals lower damage accumulation inside the joint region than CON0 at similar drift levels up to final failure.

Specimen CON0F2X has been strengthened at the joint body with X-shaped CFRP ropes. In this case, hairline cracks were observed during the first loading cycles both in the joint region and at the end of the beam. In subsequent loading cycles, severe damage was accumulated in the joint region. However, the cracks were smaller, and not as wide as those in the reference specimen CON0. Figure 8a shows the damage state of the specimen CON0F2X at the 5th step of the loading (drift 2%). The damage is further accumulated inside the joint region at the 7th step—drift 4% (Figure 8b). The case is similar for CON1F2X, revealing slightly lower damage accumulation. In both cases, the X-shaped CFRP ropes hold the fragmented joint concrete core together.

## 5. Elaboration of the P-δ Test Results

### 5.1. Envelope Curves

A thorough study of the observed maximum loads per loading step of the two specimens, and consequently of the effectiveness of the applied strengthening technique on the load carrying capacity and the ductility of the joints, can be achieved through the envelope curves as obtained from the hysteretic responses.

Comparative presentations of the envelope curves of the unstrengthened specimens (CON0 and CON1) and the strengthened specimens (CON0F2X and CON1F2X) with and without stirrups inside the joint are included in Figure 9 for the first cycle (Figure 9a), the second cycle (Figure 9b) and the third cycle (Figure 9c) of each loading step. From these comparisons, it is observed that the CFRP strengthening has increased the load carrying capacity and the ductility of the strengthened joint compared to the unstrengthened one in the high drifts of 3% and 4%. The comparative curves of specimens CON1-CON0F2X show the envelope curves almost coincide.

### 5.2. Joint Shear Deformation γ_avg_

Based on the approaches of seismic-resistant design and redesign of RC structures, the main aim for the joint region is to remain elastic in response. The variation of the shear deformation of the joint panel versus the drift ratio of all specimens has been measured to assess the damage accumulation inside the joint. The joint shear deformation has been estimated from the values of the diagonal shortening and diagonal elongation measured using two string displacement transducers diagonally mounted on the joint panel zone (Figure 10). An average value of the joint shear deformation can be calculated based on the Equation (1)
(1)$$\gamma_{avg}=\frac{\Delta l1+\Delta l2}{Lsin2\theta \;}$$
where *γ_avg_* is the average value of the joint shear deformation in rad, Δ*l1* and Δ*l2* represent the variations in the length of the strings of the diagonal string displacement transducers, *L* is the initial length of their strings and they are equal to 420 mm, *θ* is the inclination angle of the diagonals to the vertical direction and in these cases *θ* = 45.

Comparative presentation of maximum absolute values of the joint shear deformations in CON0 deviate from the elastic range during 2% lateral drift and during the 2nd cycle. In CON1 the disintegration of the joint initiates during 2% lateral drift and during the 3rd cycle. Specimen CON0F2X deviates from elastic response during 2% lateral drift during the 3rd cycle. Finally, CON1F2X maintains the elastic response up to the 1st cycle at 3% drift. Significant disintegration of the joint region for CON1F2X occurs at 4% drift, 1st cycle, for *γ_avg_* 30 × 10^−4^ rad. The drift level for similar shear deformations for CON0F2X occurs at 3% drift, 2nd cycle, for CON1 at 3% drift, 2nd cycle, and for CON0 at 2% drift, 3rd cycle. From the figures it is apparent that as the imposed displacement increases, the observed joint shear deformations of specimen CON1F2X are substantially reduced compared to the corresponding ones of the unstrengthened specimen CON1. The low deformations observed for the strengthened specimen CON1F2X are apparently attributed to the shear strengthening imposed by the X-shaped CFRP ropes externally applied on the joint body. From these observations it can be concluded that the strengthening technique improves the joints in terms of shear bearing capacity. The limit drifts for elastic joint cyclic behavior are clearly assessed. No cyclic loading effect is evidenced for CON1F2X up to 3% drift, for CON0F2X up to 2% drift, for CON1 up to 2% drift and for CON0 up to 2% drift. Specimens CON1 and CON0F2X reveal an almost identical variation of shear deformations with the beam drift for the 1st, 2nd or 3rd cycles.

## 6. Conclusions

This paper investigates the effects of inadequate internal steel stirrup quantity or of light external joint strengthening on the seismic performance of large-scale deficient T joints. The joint region is as large as 400 mm × 500 mm × 250 mm and receives X-shaped CFRP rope strengthening in an improved versatile continuous rope form with improved end anchorage detailing, The main aim is to assess the efficiency of the new detailing of the rope and to estimate low boundary adequate strengthening of rope technique for RC joints. The joints are subjected to imposed end beam cyclic displacement of increasing magnitude and three subsequent repetitions per cycle. The criteria for the assessment of the mechanical behavior of the joints are (i) the beam drift and reversal in which the post-peak shear force reduces to 80% of maximum shear (P_max_), that is at 0.8 P_max_, and (ii) the beam drift and reversal in which the shear deformation of the joint (diagonal deformation γ) exceeds the elastic range, denoting initiation of severe joint deterioration.

The following conclusions are drawn:All T specimens reveal improved and symmetric P-δ response in the presence of steel stirrups in the joint region or/and external rope strengthening when compared with as-built joints. No fracture of the rope is evidenced even for beam drifts higher than the global “failure” drifts. The proposed continuous detailing of the rope offers improved versatility and efficiency.All T specimens under investigation reach the shear force of the beam (P) that corresponds to the yielding of its tensile steel reinforcement.The higher the shear reinforcement in the form of an internal steel stirrup or/and X-shaped CFRP ropes, the higher the displacement ductility measured at the beam end at failure point of 0.8 P_max_ and the higher the number of cycles they sustain. That is, the unstrengthened joint without an internal steel stirrup CON0 fails at drift 2% during the 2nd reversal, CON1 with one internal steel stirrup as well as CON0F2X without a stirrup but with versatile X-shaped continuous CFRP rope strengthening fail at drift 3% during the 2nd reversal, and finally CON1F2X with a stirrup and CFRP rope fails at a drift close to 4% during the 1st cycle. Interestingly, CON1 and CON0F2X reveal rather equivalent mechanical response up to failure. This is extremely interesting for future redesign elaborations.The beam displacement ductility includes the contribution of the rotation of the joint based on the stiffness of the columns and of the beam, as well as the shear deformation of the joint. The results suggest that, in the absence of a stirrup in the CON0 joint, the γ values exceed the elastic range simultaneously with the 0.8 P_max_ failure point; that is, the deterioration of the joint initiates at 2% drift 2nd cycle and is, therefore, abrupt. In the presence of a steel stirrup, or alternatively of X-shaped elastic rope, the disintegration of the joint initiates at 2% drift during the 3rd cycle but it develops at a lower rate, as the shear load of the specimen is kept high up to the 2nd cycle at 3% drift. The best response is revealed for CON1F2X with elastic joint response up to the 1st cycle of 3% drift. Then, the joint starts to deteriorate while during the next cycles the load is kept high up to 4% drift.No failure of the rope strengthening is evidenced in any case.

Based on the recent development of retrofit codes (i.e., Greek or Italian) considering multiple performance levels, these tests offer significant insight into the variable behavior of real scale RC joints at different targeted drifts at 2%, 3% or 4%. It may favor the development of suitable redesign tools toward efficient and versatile retrofits of deficient joints to upgrade the displacement ductility of RC subsystems.

## Figures and Tables

**Figure 1 materials-16-02718-f001:**
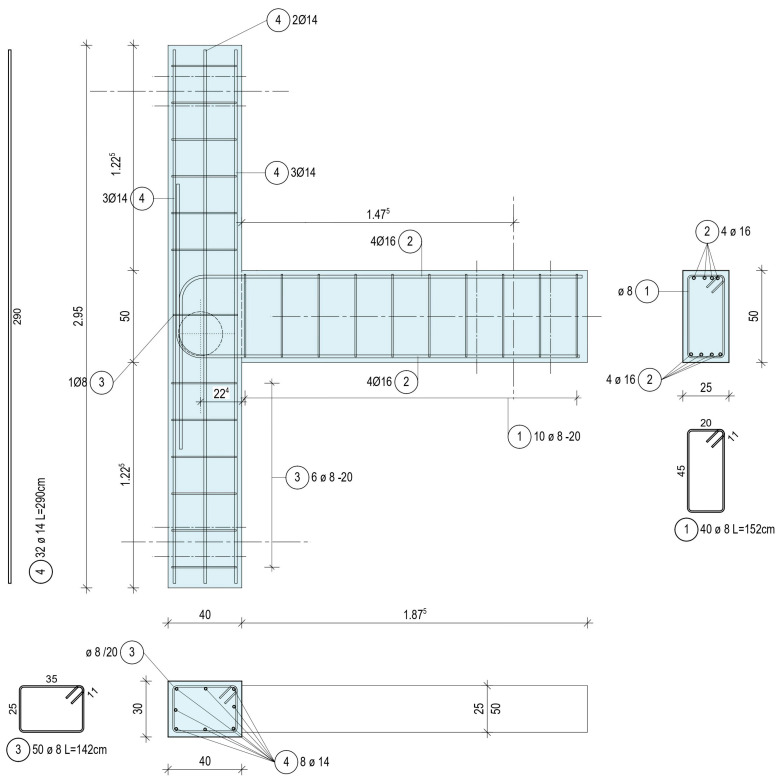
Geometry and steel reinforcement of specimen CON1 (dimensions in m).

**Figure 2 materials-16-02718-f002:**
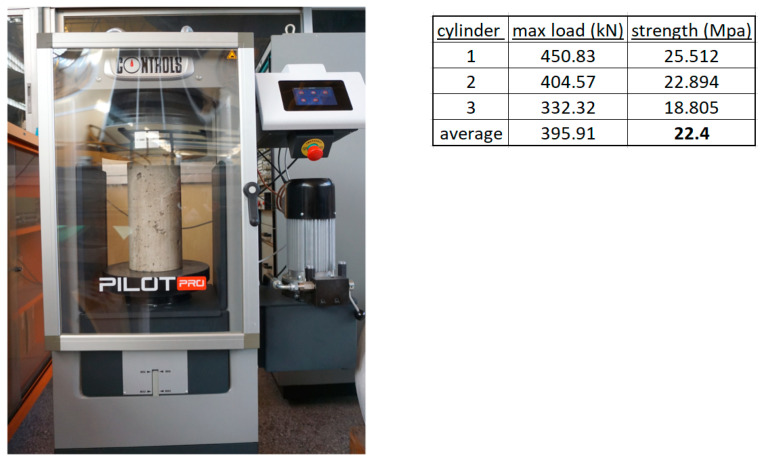
Compression tests on concrete cylinders and results.

**Figure 3 materials-16-02718-f003:**
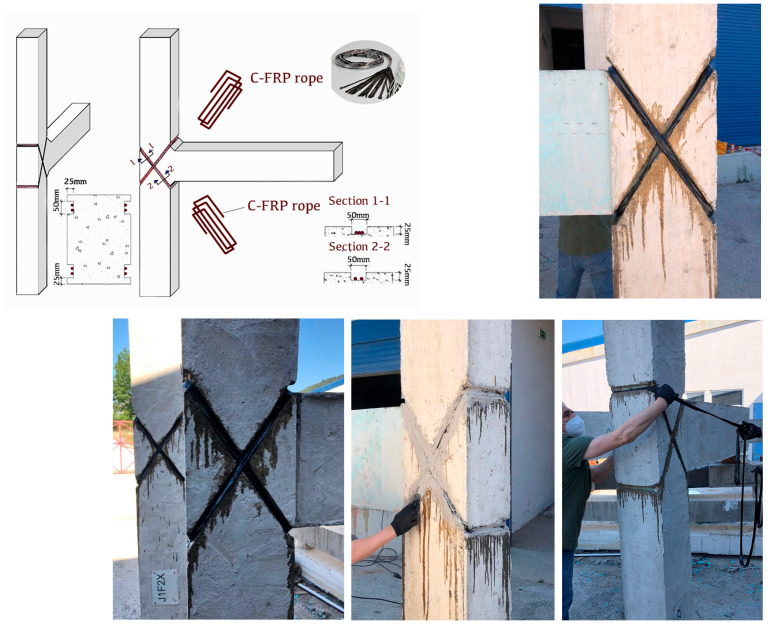
Strengthened specimens CON0F2X, CON1F2X; C-FRP ropes are applied in an X-shape form on the joint body.

**Figure 4 materials-16-02718-f004:**
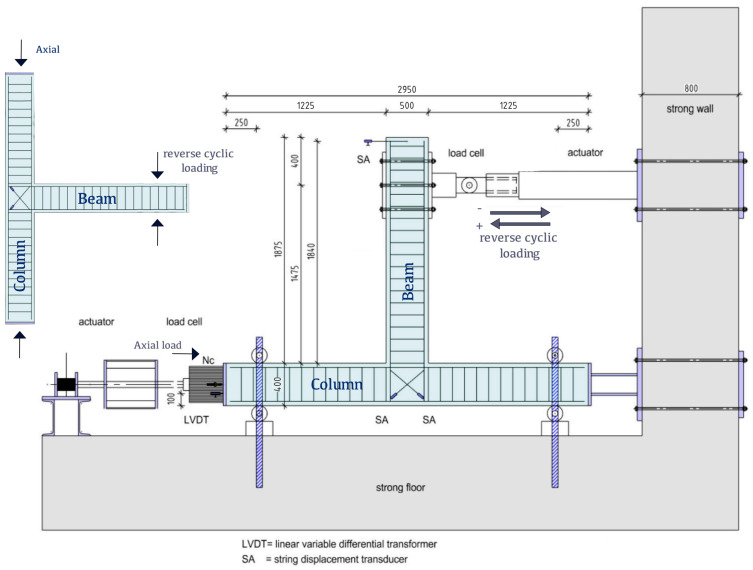
Test setup and instrumentation.

**Figure 5 materials-16-02718-f005:**
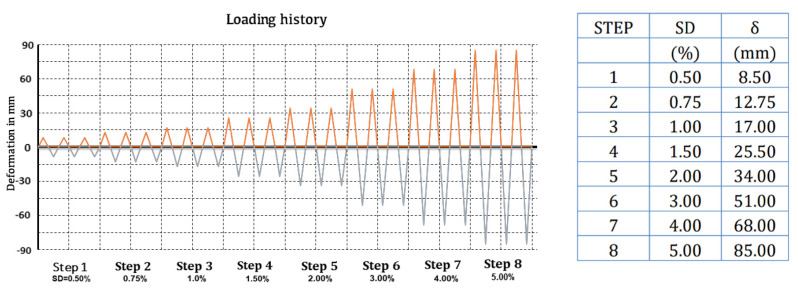
Loading sequence, drift ratios and corresponding displacements.

**Figure 6 materials-16-02718-f006:**
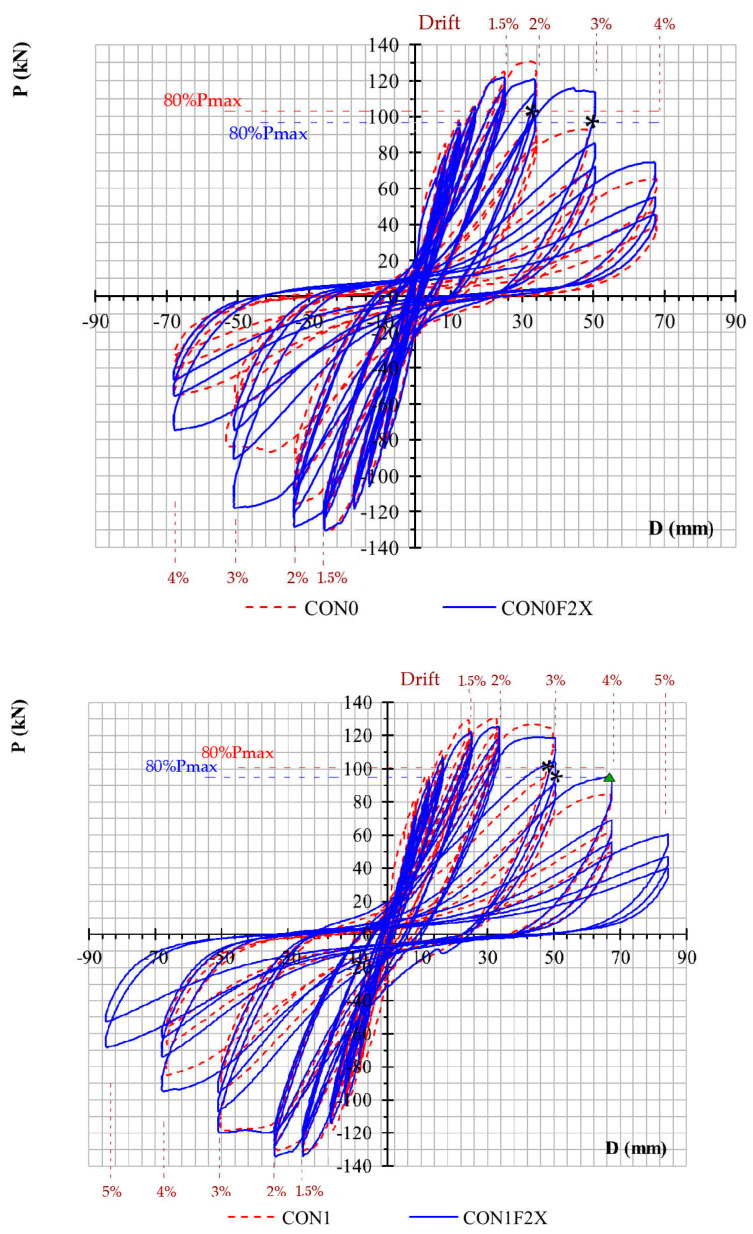
Comparative presentation of the hysteretic responses of the specimens.

**Figure 7 materials-16-02718-f007:**
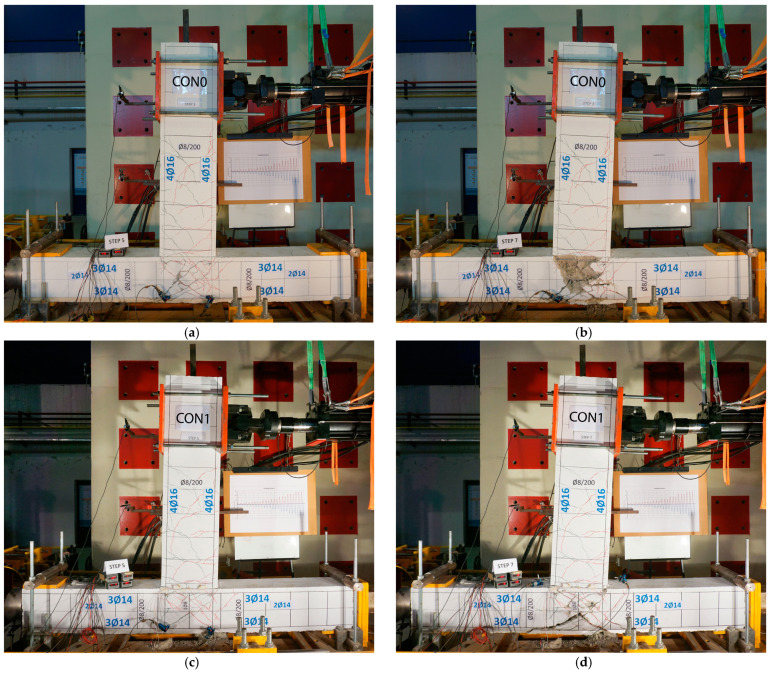
Damage states of unstrengthened specimens. (**a**) Specimen CON0. Damage state at the 5th step (drift 2%). (**b**) Specimen CON0. Damage state at the 7th step (drift 4%). (**c**) Specimen CON1. Damage state at the 5th step (drift 2%). (**d**) Specimen CON1. Damage state at the 7th step (drift 4%).

**Figure 8 materials-16-02718-f008:**
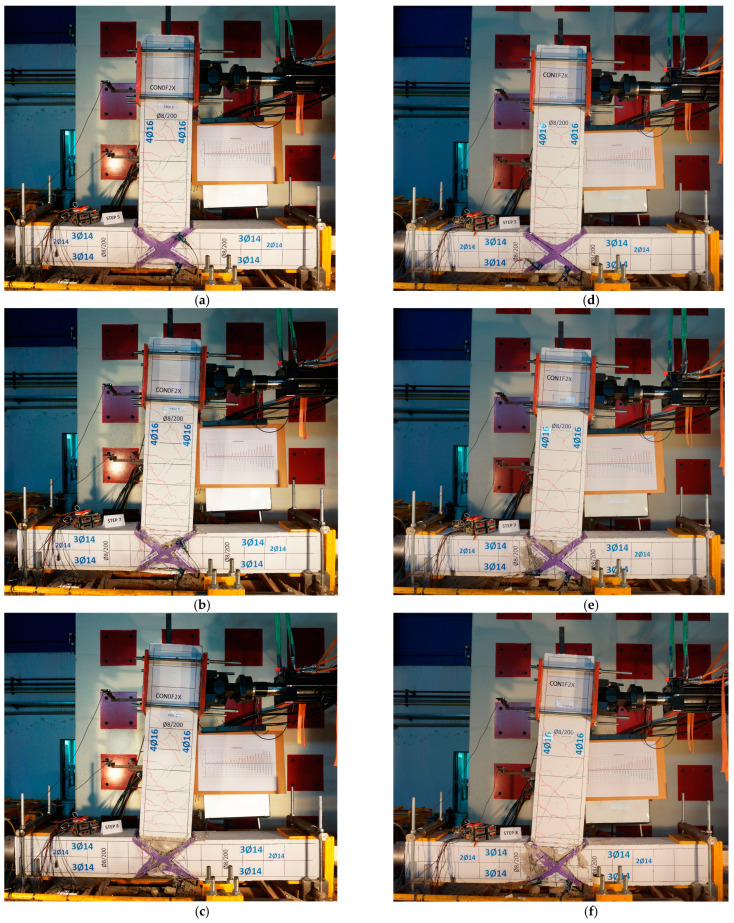
Damage states of strengthened specimens. (**a**) Specimen CON0F2X. Damage state at the 5th step (drift 2%). (**b**) Specimen CON0F2X. Damage state at the 7th step (drift 4%). (**c**) Specimen CON0F2X. Damage state at the 8th step (drift 5%). (**d**) Specimen CON1F2X. Damage state at the 5th step (drift 2%). (**e**) Specimen CON1F2X. Damage state at the 7th step (drift 4%). (**f**) Specimen CON1F2X. Damage state at the 8th step (drift 5%).

**Figure 9 materials-16-02718-f009:**
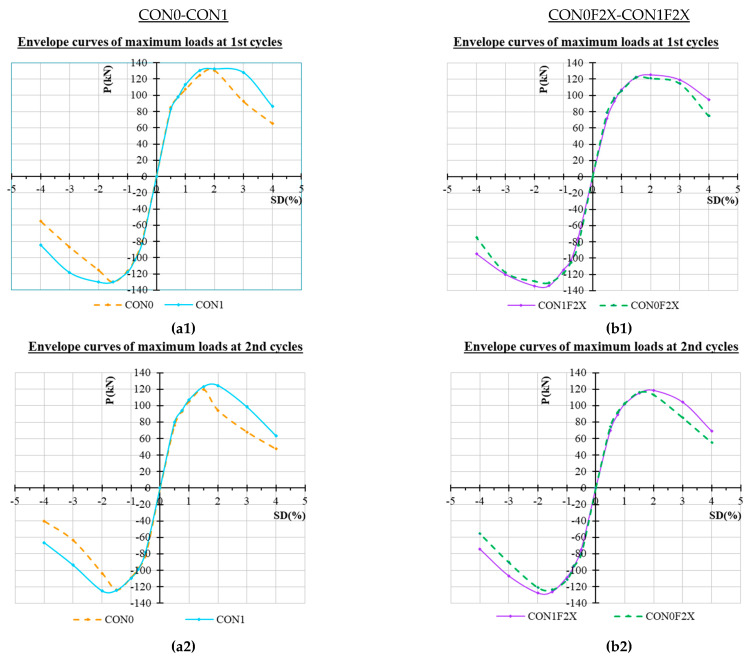
Comparative presentation of the envelope curves of the hysteretic responses of the specimens. (**a1**–**a3**) Maximum loads of 1st, 2nd and 3rd cycles of unstrengthened specimens. (**b1**–**b3**) Maximum loads of 1st, 2nd and 3rd cycles of strengthened specimens.

**Figure 10 materials-16-02718-f010:**
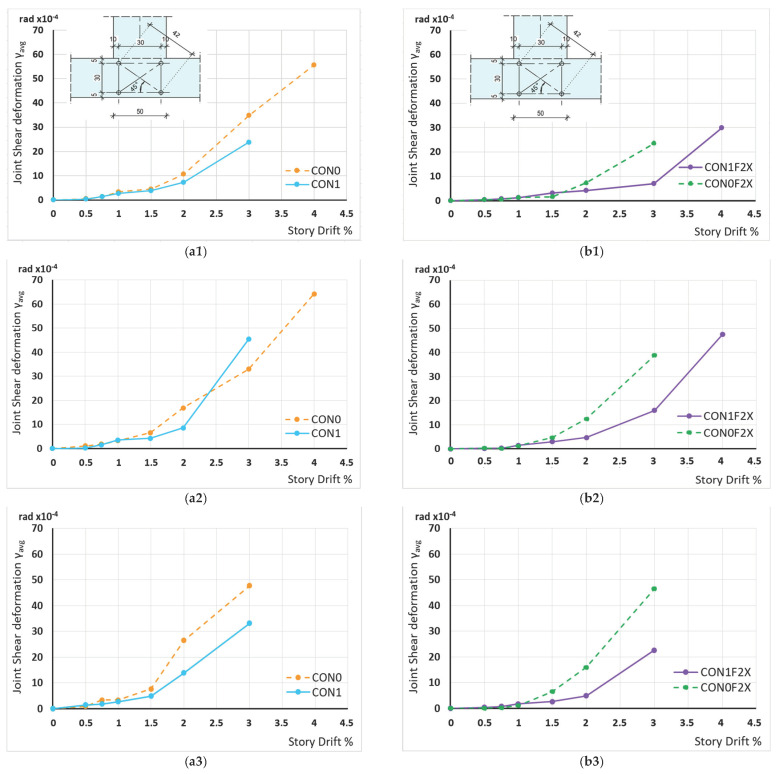
Shear deformation of CON0-CON1 (**a**) and CON0F2X-CON1F2X (**b**) at (**a1**,**b1**) 1st cycle, (**a2**,**b2**) 2nd cycle, (**a3**,**b3**) 3rd cycle. Shear deformations of the specimens’ joint body as obtained by the diagonally mounted string displacement transducers (LVDTs).

## Data Availability

Data available on request.

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
