# Peer review of "Seismic Tests of Full Scale Reinforced Concrete T Joints with Light External Continuous Composite Rope Strengthening—Joint Deterioration and Failure Assessment"

_materials, 2023, doi:10.3390/ma16072718_

Round 1

Reviewer 1 Report

In this manuscript, the cyclic behavior of RC T joints strengthened with light external continuous composite rope were investigated experimentally. The issue is interesting and the manuscript is well written. Some suggestions and questions are provided below to be applied and answered before accepting the manuscript:

-       What is the reason behind choosing near surface method (NSM) with applying grooves for applying the composite ropes to the T joints? As the grooves may cause decrease in concrete strength, is there no way to apply the ropes externally without any groove?

-       The abstract can be summarized and report more quantitative results.

-       The resolution of Figures 1, 4, and 6 can be improved.

-       It is suggested to add the energy absorption of T joints as a comparing parameter.

-       There are some other related publications which can be added to the literature review. Some of the related publications provided below:

·         Obaidat, Y. T. (2022, March). Cyclic behavior of interior RC beam-column joints strengthened with NSM-CFRP ropes. In Structures (Vol. 37, pp. 735-744). Elsevier.

·         Jahangir, H., Rezazadeh Eidgahee, D., & Esfahani, M. R. (2022). Bond strength characterization and estimation of steel fibre reinforced polymer-concrete composites. Steel and Composite Structures44.

·         Saeed, Y. M., Aules, W. A., & Rad, F. N. (2022). Flexural strengthening of RC columns with EB-CFRP sheets and NSM-CFRP rods and ropes. Composite Structures301, 116236.

-       The conclusion section can be summarized and presented as bullet format.

Reviewer 2 Report

The paper ((Seismic tests of full scale reinforced concrete T joints with light external continuous composite rope strengthening – Joint deterioration and failure assessment)) is a good paper and suitable to publish in the materials journal after treating the comments below:

1.      Please try to correct the grammatical errors in the abstract section. 

2.      Please correct the grammatical errors in the introduction part. 

3.      Try to insert new citations not more than 10 years ago, and try to insert suitable references for your arguments.

4.      In the first paragraph of introduction part, please rewrite the sentence ((Seismic protection of RC joints includes world-wide known retrofit techniques such as traditional RC jacketing, steel plate or steel rod jacketing and advanced shear strengthening with fiber reinforced
polymer (FRP) sheets)). And what do you mean about the point and blanket in the initial sentence?

5.      In the sentence ((Externally bonded composite materials have been extensively used in different configurations in strengthening of RC columns and beams as well as in joints)). The authors insert about 15 references for this argument, and this is not suitable, while there are other arguments without any citation???!!! So, please summarize and use the newest and related to this topic.

6.      It is better to insert a new table for the abbreviations and symbols used in this study.

7.      Fig. 8 did not mention in the text, so please check it, carefully.

8.      Try to explain more about Figs. 6 and 7.

9.      You need to discuss your results in more detail.

10.  In the conclusion part; You cannot insert any citation in the conclusion part. You should present your results scientifically and try to insert some numerical results. It is better to mention some recommendations for future studies.

Round 2

Reviewer 1 Report

The revised version applied all suggestions.

Reviewer 2 Report

Accept